# GumBolt: Extending Gumbel trick to Boltzmann priors

**Amir H. Khoshaman**
D-Wave Systems Inc.*
khoshaman@gmail.com

**Mohammad H. Amin**
D-Wave Systems Inc.
Simon Fraser University
mhsamin@dwavesys.com

## Abstract

Boltzmann machines (BMs) are appealing candidates for powerful priors in variational autoencoders (VAEs), as they are capable of capturing nontrivial and multimodal distributions over discrete variables. However, non-differentiability of the discrete units prohibits using the reparameterization trick, essential for low-noise back propagation. The Gumbel trick resolves this problem in a consistent way by relaxing the variables and distributions, but it is incompatible with BM priors. Here, we propose the GumBolt, a model that extends the Gumbel trick to BM priors in VAEs. GumBolt is significantly simpler than the recently proposed methods with BM prior and outperforms them by a considerable margin. It achieves state-of-the-art performance on permutation invariant MNIST and OMNIGLOT datasets in the scope of models with only discrete latent variables. Moreover, the performance can be further improved by allowing multi-sampled (importance-weighted) estimation of log-likelihood in training, which was not possible with previous models.

## 1 Introduction

Variational autoencoders (VAEs) are generative models with the useful feature of learning representations of input data in their latent space. A VAE comprises of a prior (the probability distribution of the latent space), a decoder and an encoder (also referred to as the approximating posterior or the inference network). There have been efforts devoted to making each of these components more powerful. The decoder can be made richer by using autoregressive methods such as pixelCNNs, pixelRNNs (Oord et al., 2016) and MADEs (Germain et al., 2015). However, VAEs tend to ignore the latent code (in the sense described by Yeung et al. (2017)) in the presence of powerful decoders (Chen et al., 2016; Gulrajani et al., 2016; Goyal et al., 2017). There are also a myriad of works strengthening the encoder distribution (Kingma et al., 2016; Rezende and Mohamed, 2015; Salimans et al., 2015). Improving the priors is manifestly appealing, since it directly translates into a more powerful generative model. Moreover, a rich structure in the latent space is one of the main purposes of VAEs. Chen et al. (2016) observed that a more powerful autoregressive prior and a simple encoder is commensurate with a powerful inverse autoregressive approximating posterior and a simple prior.

Boltzmann machines (BMs) are known to represent intractable and multi-modal distributions (Le Roux and Bengio, 2008), ideal for priors in VAEs, since they can lead to a more expressive generative model. However, BMs contain discrete variables which are incompatible with the reparameterization trick, required for efficient propagation of gradients through stochastic units. It is desirable to have discrete latent variables in many applications such as semi-supervised learning

(Kingma et al., 2014), semi-supervised generation (Maaløe et al., 2017) and hard attention models (Serrà et al., 2018; Gregor et al., 2015), to name a few. Many operations, such as choosing between models or variables are naturally expressed using discrete variables (Yeung et al., 2017).

Rolfe (2016) proposed the first model to use a BM in the prior of a VAE, i.e., a discrete VAE (dVAE). The main idea is to introduce auxiliary continuous variables (Fig. 1(a)) for each discrete variable through a "smoothing distribution". The discrete variables are marginalized out in the autoencoding term by imposing certain constraints on the form of the relaxing distribution. However, the discrete variables cannot be marginalized out from the remaining term in the objective (the KL term). Their proposed approach relies on properties of the smoothing distribution to evaluate these terms. In Appendix B, we show that this approach is equivalent to REINFORCE when dealing with some parts of the KL term (*i.e.*, the cross-entropy term). Vahdat et al. (2018) proposed an improved version, dVAE++, that uses a modified distribution for the smoothing variables, but has the same form for the autoencoding part (see Sec. 2.1). The qVAE, (Khoshaman et al., 2018), expanded the dVAE to operate with a quantum Boltzmann machine (QBM) prior (Amin et al., 2016). A major shortcoming of these methods is that they are unable to have multi-sampled (importance-weighted) estimates of the objective function during training, which can improve the performance.

To use the reparameterization trick directly with discrete variables (without marginalization), a continuous and differentiable proxy is required. The Gumbel (reparameterization) trick, independently developed by Jang et al. (2016) and Maddison et al. (2016), achieves this by relaxing discrete distributions. However, BMs and in general discrete random Markov fields (MRFs) are incompatible with this method. Relaxation of discrete variables (rather than distributions) for the case of factorial categorical prior (Gumbel-Softmax) was also investigated in both works. It is not obvious whether such relaxation of discrete variables would work with BM priors.

The contributions of this work are as follows: we propose the GumBolt, which extends the Gumbel trick to BM and MRF priors and is significantly simpler than previous models that marginalize discrete variables. We show that BMs are compatible with relaxation of discrete variables (rather than distributions) in Gumbel trick. We propose an objective using such relaxation and show that the main limitations of previous models with BM priors can be circumvented; we do not need marginalization of the discrete variables, and can have an importance-weighted objective. GumBolt considerably outperforms the previous works in a wide series of experiments on permutation invariant MNIST and OMNIGLOT datasets, even without the importance-weighted objective (Sec. 5). Increasing the number of importance weights can further improve the performance. We obtain the state-of-the-art results on these datasets among models with only discrete latent variables.

## 2 Background

### 2.1 Variational autoencoders

Consider a generative model involving observable variables $x$ and latent variables $z$. The joint probability distribution can be decomposed as $p_\theta(x, z) = p_\theta(z)p_\theta(x|z)$. The first and second terms on the right hand side are the prior and decoder distributions, respectively, which are parametrized by $\theta$. Calculating the marginal $p_\theta(x)$ involves performing intractable, high dimensional sums or integrals. Assume an element $x$ of the dataset $\mathcal{X}$ comprising of $N$ independent samples from an unknown underlying distribution is given. VAEs operate by introducing a family of approximating posteriors $q_\phi(z|x)$ and maximize a lower bound (also known as the ELBO), $\mathcal{L}(x; \theta, \phi)$, on the log-likelihood $\log p_\theta(x)$ (Kingma and Welling, 2013):

$$\log p_\theta(x) \geq \mathcal{L}(x; \theta, \phi) = \mathop{\mathbb{E}}_{q_\phi(z|x)} \left[ \log \frac{p_\theta(x, z)}{q_\phi(z|x)} \right]$$
$$= \mathop{\mathbb{E}}_{q_\phi(z|x)} \left[ \log p_\theta(x|z) \right] - D_{KL}\big(q_\phi(z|x) \,\|\, p_\theta(z)\big), \tag{1}$$

where the first term on the right-hand side is the autoencoding term and $D_{\text{KL}}$ is the Kullback-Leibler divergence (Bishop, 2011). In VAEs, the parameters of the distributions (such as the means in the case of Bernoulli variables) are calculated using neural nets. To backpropagate through latent variables $z$, the reparameterization trick is used; $z$ is reparametrized as a deterministic function $f(\phi, x, \rho)$, where the stochasticity of $z$ is relegated to another random variable, $\rho$, from a distribution that does not depend on $\phi$. Note that it is impossible to backpropagate if $z$ is discrete, since $f$ is not differentiable.

## 2.2 Gumbel trick

The non-differentiability of $f$ can be resolved by finding a relaxed proxy for the discrete variables. Assume a binary unit, $z$, with mean $\bar{q}$ and logit $l$; *i.e.*, $p(z=1) = \bar{q} = \sigma(l)$, where $\sigma(l) \equiv \frac{1}{1+\exp(-l)}$ is the sigmoid function. Since $\sigma(l)$ is a monotonic function, we can reparametrize $z$ as $z = \mathcal{H}(\rho - (1-\bar{q})) = \mathcal{H}\left(l + \sigma^{-1}(\rho)\right)$ (Maddison et al., 2016), where $\mathcal{H}$ is the Heaviside function, $\rho \sim \mathcal{U}$ with $\mathcal{U}$ being a uniform distribution in the range $[0,1]$, and $\sigma^{-1}(\rho) = \log(\rho) - \log(1-\rho)$ is the inverse sigmoid or logit function. This transformation results in a non-differentiable reparameterization, but can be smoothed when the Heaviside function is replaced by a sigmoid function with a temperature $\tau$, *i.e.*, $\mathcal{H}(\ldots) \to \sigma(\frac{\ldots}{\tau})$. Thus, we introduce the continuous proxy (Maddison et al., 2016):

$$f(\phi, x, \rho) = \zeta = \sigma\left(\frac{l(\phi, x) + \sigma^{-1}(\rho)}{\tau}\right). \tag{2}$$

The continuous $\zeta$ is differentiable and is equal to the discrete $z$ in the limit $\tau \to 0$.

## 2.3 Boltzmann machine

Our goal is to use a BM as prior. A BM is a probabilistic energy model described by

$$p_\theta(z) = \frac{e^{-E_\theta(z)}}{Z_\theta}, \tag{3}$$

where $E_\theta(z)$ is the energy function, and $Z_\theta = \sum_{\{z\}} e^{-E_\theta(z)}$ is the partition function; $z$ is a vector of binary variables. Since finding $p_\theta(z)$ is typically intractable, it is common to use sampling techniques to estimate the gradients. To facilitate MCMC sampling using Gibbs-block technique, the connectivity of latent variables is assumed to be bipartite; *i.e.*, $z$ is decomposed as $[z_1, z_2]$ giving

$$-E_\theta(z) = a^T z_1 + b^T z_2 + z_2^T W z_1, \tag{4}$$

where $a$, $b$ and $W$ are the biases (on $z_1$ and $z_2$, respectively) and weights. This bipartite structure is known as the restricted Boltzmann Machine.

# 3 Proposed approach

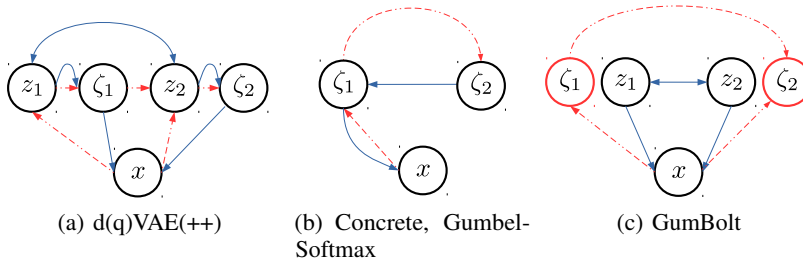

(a) d(q)VAE(++)&emsp;&emsp;(b) Concrete, Gumbel-Softmax&emsp;&emsp;(c) GumBolt

Figure 1: Schematic of the discussed models with discrete variables in their latent space. The dashed red and solid blue arrows represent the inference network, and the generative model, respectively. (a) dVAE, qVAE (Khoshaman et al., 2018) and dVAE++ have the same structure. They involve auxiliary continuous variables, $\zeta$, for each discrete variable, $z$, provided by the same conditional probability distribution, $r(\zeta|z)$, in both the generative and approximating posterior networks. (b) Concrete and Gumbel-Softmax apply the Gumbel trick to the discrete variables to obtain the $\zeta$s that appear in both the inference and generative models. (c) GumBolt only involves discrete variables in the generative model, and the relaxed $\zeta$s are used in the inference model during training. Note that during evaluation, the temperature is set to zero, leading to $\zeta = z$.

The importance-weighted or multi-sampled objective of a VAE with BM prior can be written as:

$$\log p_\theta(x) \geq \mathcal{L}_k(x; \theta, \phi) = \underset{\prod_i q_\phi(z^i|x)}{\mathbb{E}} \left[ \log \frac{1}{k} \sum_{i=1}^k \frac{p_\theta(z^i)p_\theta(x|z^i)}{q_\phi(z^i|x)} \right]$$

$$= \underset{\prod_i q_\phi(z^i|x)}{\mathbb{E}} \left[ \log \frac{1}{k} \sum_{i=1}^k \frac{e^{-E_\theta(z^i)}p_\theta(x|z^i)}{q_\phi(z^i|x)} \right] - \log Z_\theta,$$

(5)

where $k$ is the number of samples or importance-weights over which the Monte Carlo objective is calculated, (Mnih and Rezende, 2016). $z^i$ are independent vectors sampled from $q_\phi(z^i|x)$. Note that we have taken out $Z_\theta$ from the argument of the expectation value, since it is independent of $z$. The partition function is intractable but its derivative can be estimated using sampling:

$$\nabla_\theta \log Z_\theta = \nabla_\theta \log \sum_{\{z\}} e^{-E_\theta(z)} = -\frac{\sum_{\{z\}} \nabla_\theta E_\theta(z) e^{-E_\theta(z)}}{\sum_{\{z\}} e^{-E_\theta(z)}} = -\underset{p_\theta(z)}{\mathbb{E}} \left[ \nabla_\theta E_\theta(z) \right]. \quad (6)$$

Here, $\sum_{\{z\}}$ involves summing over all possible configurations of the binary vector $z$. The objective $\mathcal{L}_k(x; \theta, \phi)$ cannot be used for training, since it involves non-differentiable discrete variables. This can be resolved by relaxing the distributions:

$$\log p_\theta(x) \geq \tilde{\mathcal{L}}_k(x; \theta, \phi) = \underset{\prod_i q_\phi(\zeta^i|x)}{\mathbb{E}} \left[ \log \frac{1}{k} \sum_{i=1}^k \frac{p_\theta(\zeta^i)p_\theta(x|\zeta^i)}{q_\phi(\zeta^i|x)} \right]$$

$$= \underset{\prod_i q_\phi(\zeta^i|x)}{\mathbb{E}} \left[ \log \frac{1}{k} \sum_{i=1}^k \frac{e^{-E_\theta(\zeta^i)}p_\theta(x|\zeta^i)}{q_\phi(\zeta^i|x)} \right] - \log \tilde{Z}_\theta.$$

Here, $\zeta^i$ is a continuous variable sampled from Eq. 2, which is consistent with the Gumbel probability $q_\phi(\zeta^i|x)$ defined in (Maddison et al., 2016), and $p_\theta(\zeta) \equiv e^{-E_\theta(\zeta)}/\tilde{Z}_\theta$, where $\tilde{Z}_\theta \equiv \int d\zeta e^{-E_\theta(\zeta)}$. The expectation distribution is the joint distribution over independent $z^i$ samples. Notice that $\log \tilde{Z}_\theta$ is different from $\log Z_\theta$, therefore its derivatives cannot be estimated using discrete samples from a BM, making this method inapplicable for BM priors. The derivatives could be estimated using samples from a continuous distribution, which is very different from the BM distribution. Analytical calculation of the expectations, suggested for Bernoulli prior by Maddison et al. (2016) is also infeasible for BMs, since it requires exhaustively summing over all possible configurations of the binary units.

## 3.1 GumBolt probability proxy

To replace $\log \tilde{Z}_\theta$ with $\log Z_\theta$, we introduce a proxy probability distribution:

$$\breve{p}_\theta(\zeta) \equiv \frac{e^{-E_\theta(\zeta)}}{Z_\theta}. \quad (7)$$

Note that $\breve{p}_\theta(\zeta)$ is not a true (normalized) probability density function, but $\breve{p}_\theta(\zeta) \rightarrow p_\theta(z)$ as $\tau \rightarrow 0$.

Now consider the following theorems (see Appendix A for proof):

**Theorem 1.** *For any polynomial function $E_\theta(z)$ of $n_z$ binary variables $z \in \{0,1\}^{n_z}$, the extrema of the relaxed function $E_\theta(\zeta)$ with $\zeta \in [0,1]^{n_z}$ reside on the vertices of the hypercube, i.e., $\zeta_{\text{extr}} \in \{0,1\}^{n_z}$.*

**Theorem 2.** *For any polynomial function $E_\theta(z)$ of $n_z$ binary variables $z \in \{0,1\}^{n_z}$, the proxy probability $\breve{p}_\theta(\zeta) \equiv e^{-E_\theta(\zeta)}/Z_\theta$, with $\zeta \in [0,1]^{n_z}$, is a lower bound to the true probability $p_\theta(\zeta) \equiv e^{-E_\theta(\zeta)}/\tilde{Z}_\theta$, i.e., $\breve{p}_\theta(\zeta) \leq p_\theta(\zeta)$, where $Z_\theta \equiv \sum_{\{z\}} e^{-E_\theta(z)}$ and $\tilde{Z}_\theta \equiv \int_{\{\zeta\}} d\zeta e^{-E_\theta(\zeta)}$.*

Therefore, according to theorem (2), replacing $p_\theta(\zeta)$ with $\breve{p}_\theta(\zeta)$, we obtain a lower bound on $\tilde{\mathcal{L}}_k(x; \theta, \phi)$:

$$\breve{\mathcal{L}}_k(x; \theta, \phi) = \underset{\prod_i q_\phi(\zeta^i|x)}{\mathbb{E}} \left[ \log \frac{1}{k} \sum_{i=1}^k \frac{e^{-E_\theta(\zeta^i)}p_\theta(x|\zeta^i)}{q_\phi(\zeta^i|x)} \right] - \log Z_\theta \leq \tilde{\mathcal{L}}_k(x; \theta, \phi). \quad (8)$$

This allows reparameterization trick, while making it possible to use sampling to estimate the gradients. The structure of our model with a BM prior is portrayed in Figure 1(c), where both continuous and discrete variables are used. Notice that in the limit $\tau \to 0$, $\breve{p}_\theta(\zeta^i)$ becomes a probability mass function (pmf) $p_\theta(z^i)$, while $q_\phi(\zeta^i|x)$ remains as a probability density function (pdf). To resolve this inconsistency, we replace $q_\phi(\zeta^i|x)$ with the Bernoulli pmf: $\breve{q}_\phi(\zeta^i|x) = \zeta^i \log \bar{q}^i + (1 - \zeta^i) \log(1 - \bar{q}^i)$, which approaches $q_\phi(z^i|x)$ when $\tau \to 0$. The training objective

$$\mathcal{F}_k(x; \theta, \phi) = \mathbb{E}_{\prod_i q_\phi(\zeta^i|x)} \left[ \log \frac{1}{k} \sum_{i=1}^{k} \frac{e^{-E_\theta(\zeta^i)} p_\theta(x|\zeta^i)}{\breve{q}_\phi(\zeta^i|x)} \right] - \log Z_\theta \tag{9}$$

becomes $\mathcal{L}_k(x; \theta, \phi)$ at $\tau = 0$ as desired (see Fig. 2(a) for the relationship among different objectives). This is the analog of the Gumbel-Softmax trick (Jang et al., 2016) when applied to BMs. During training, $\tau$ should be kept small for the continuous variables to stay close to the discrete variables, a common practice with Gumbel relaxation (Tucker et al., 2017). For evaluation, $\tau$ is set to zero, leading to an unbiased evaluation of the objective function. For generation, discrete samples from BM are directly fed into the decoder to obtain the probabilities of each input feature.

## 3.2 Compatibility of BM with discrete variable relaxation

The term in the discrete objective that involves the prior distribution is $\mathbb{E}_{q_\phi(z|x)} [\log p_\theta(z)]$. When $z$ is replaced with $\zeta$, there is no guarantee that the parameters that optimize $\mathbb{E}_{q_\phi(\zeta|x)} [\log p_\theta(\zeta)]$ would also optimize the discrete version. This happens naturally for Bernoulli distribution used in (Jang et al., 2016), since the extrema of the prior term in the objective $\log(p(\zeta)) = \zeta \log(\bar{p}) + (1 - \zeta) \log(1 - \bar{p})$ occur at the boundaries (*i.e.*, when $\zeta = 1$ or $\zeta = 0$). This means that throughout the training, the values of $\zeta$ are pushed towards the boundary points, consistent with the discrete objective.

In the case of a BM prior, according to theorem (1) (proved in Appendix A), the extrema of $\log p_\theta(\zeta) \propto -E_\theta(z)$ also occur on the boundaries; this shows that having a BM rather than a factorial Bernoulli distribution does not exacerbate the training of GumBolt.

## 4 Related works

Several approaches have been devised to calculate the derivative of the expectation of a function with respect to the parameters of a Bernoulli distribution, $\mathcal{I} \equiv \nabla_\phi \mathbb{E}_{q_\phi(z)} [f(z)]$:

1. Analytical method: for simple functions, *e.g.*, $f(z) = z$, one can analytically calculate the expectation and obtain $\mathcal{I} = \nabla_\phi \mathbb{E}_{q_\phi(z)} [z] = \nabla_\phi \bar{q}$, where $\bar{q}$ is the mean of the Bernoulli distribution. This is a non-biased estimator with zero variance, but can only be applied to very simple functions. This approach is frequently used in semi-supervised learning (Kingma et al., 2014) by summing over different categories.

2. Straight-through method: continuous proxies are used in backpropagation to evaluate derivatives, but discrete units are used in forward propagation (Bengio et al., 2013; Raiko et al., 2014).

3. REINFORCE trick: $\mathcal{I} = \mathbb{E}_{q_\phi(z)} [f(z) \nabla_\phi \log q_\phi(z)]$, although it has high variance, which can be reduced by variance reduction techniques (Williams, 1992).

4. Reparameterization trick: this method, as delineated in Sec. 2.1-2.2, is biased except in the limit where the proxies approach the discrete variables.

5. Marginalization: if possible, one can marginalize the discrete variables out from some parts of the loss function (Rolfe, 2016).

NVIL (Mnih and Gregor, 2014) and its importance-weighted successor VIMCO (Mnih and Rezende, 2016) use (3) with input-dependent signals obtained from neural networks to subtract from a baseline in order to reduce the variance of the estimator. REBAR (Tucker et al., 2017) and its generalization, RELAX (Grathwohl et al., 2017) use (3) and employ (4) in their control variates obtained using the Gumbel trick. DARN (Gregor et al., 2013) and MuProp (Gu et al., 2015) apply the Taylor expansion of the function $f(z)$ to synthesize baselines. dVAE and dVAE++ (Fig. 1(a)), which are the only works with BM priors, operate primarily based on (5) in their autoencoding term and use a combination

Table 1: Test-set log-likelihood of the GumBolt compared against dVAE and dVAE++. $k$ represents the number of samples used to calculate the objective during *training*. Note that dVAE and dVAE++ are only consistent with $k = 1$. See the main text for more details.

|  |  | dVAE | dVAE++ | GumBolt | | |
|---|---|---|---|---|---|---|
|  |  | $k = 1$ | $k = 1$ | $k = 1$ | $k = 5$ | $k = 20$ |
| **MNIST** | $-$ | 90.11 | 90.40 | 88.88 | 88.18 | **87.45** |
|  | $\sim$ | 85.72 | 85.41 | 84.86 | 84.31 | **83.87** |
|  | $--$ | 85.71 | 87.35 | 85.42 | 84.65 | **84.46** |
|  | $\sim\sim$ | 84.33 | 84.75 | 83.28 | 83.01 | **82.75** |
| **OMNIGLOT** | $-$ | 106.83 | 106.01 | 105.00 | 103.99 | **103.69** |
|  | $\sim$ | 102.85 | 101.97 | 101.61 | 101.02 | **100.68** |
|  | $--$ | 101.98 | 102.62 | 100.62 | 99.38 | **99.36** |
|  | $\sim\sim$ | 101.75 | 100.70 | 99.82 | 99.32 | **98.81** |

of (1-4) for their KL term. In Appendix B, we show that dVAE has elements of REINFORCE in calculating the derivative of the KL term. Our approach, GumBolt, exploits (4), and does not require marginalizing out the discrete units.

## 5 Experiments

In order to explore the effectiveness of the GumBolt, we present the results of a wide set of experiments conducted on standard feed-forward structures that have been used to study models with discrete latent variables (Maddison et al., 2016; Tucker et al., 2017; Vahdat et al., 2018). At first, we evaluate GumBolt against dVAE and dVAE++ baselines, all in the same framework and structure. We also demonstrate empirically that the GumBolt objective, Eq. 9, faithfully follows the non-differentiable discrete objective throughout the training. We then note on the relation between our model and other models that involve discrete variables. We also gauge the performance advantage GumBolt obtains from the BM by removing the couplings of the BM and re-evaluating the model.

### 5.1 Comparison against dVAE and dVAE++

We compare the models on statically binarized MNIST (Salakhutdinov and Murray, 2008) and OMNIGLOT datasets (Lake et al., 2015) with the usual compartmentalization into the training, validation, and test-sets. The 4000-sample estimation of log-likelihood (Burda et al., 2015) of the models are reported in Table 1. The structures used are the same as those of (Vahdat et al., 2018), which were in turn adopted from (Tucker et al., 2017) and (Maddison et al., 2016). We performed experiments with dVAE, dVAE++ and GumBolt on the same structure, and set the temperature to zero during evaluation (the results reported in (Vahdat et al., 2018) are calculated using non-zero temperatures). The inference network is chosen to be either factorial or have two hierarchies (Fig. 1(c)). In the case of two hierarchies, we have:

$$q_\phi(z|x) = q_\phi(z_1, z_2|x) = q_\phi(z_1|x)q_\phi(z_2|z_1, x)$$

, where $z = [z_1, z_2]$.

The meaning of the symbols in Table 1 are as follows: $-$, and $\sim$ represent linear and nonlinear layers in the encoder and decoder neural networks. The number of stochastic layers (hierarchies) in the encoder is equal to the number of symbols. The dimensionality of the latent space is 200 times the number of symbols; *e.g.*, $\sim\sim$ means two stochastic layers (just as in Fig. 1(c)), with 2 hidden layers (each one containing 200 deterministic units) in the encoder. The dimensionality of each stochastic layer is equal to 200 in the encoder network; the generative network is a $200 \times 200$ RBM (a total of 400 stochastic units), for $\sim\sim$ and $--$, whereas, for $-$ and $\sim$, it is a $100 \times 100$ RBM. Note that in the case of $--$, only one layer of 200 deterministic units is used in each one of the two hierarchies. The decoder network receives the samples from the RBM and probabilistically maps them into the input space using one or two layers of deterministic units. Since the RBM has bipartite structure, our model has two stochastic layers in the generative model. The chosen hyper-parameters are as follows: $1M$ iterations of parameter updates using the ADAM algorithm (Kingma and Ba, 2014),

with the default settings and batch size of 100 were carried out. The initial learning rate is $3 \times 10^{-3}$ and is subsequently reduced by 0.3 at 60%, 75%, and 95% of the total iterations. KL annealing (Sønderby et al., 2016) was used via a linear schedule during 30% of the total iterations. The value of temperature, $\tau$ was set to $\frac{1}{7}$ for all the experiments involving GumBolt, $\frac{1}{5}$ for experiments with dVAE, and $\frac{1}{10}$ and $\frac{1}{8}$ for dVAE++ on the MNIST and OMNIGLOT datasets, respectively; these values were cross-validated from $\{\frac{1}{10}, \frac{1}{9} \dots, \frac{1}{5}\}$. The GumBolt shows the same average performance for temperatures in the range $\{\frac{1}{9}, \frac{1}{8}, \frac{1}{7}\}$. The reported results are the averages from performing the experiments 5 times. The standard deviations in all cases are less than 0.15; we avoid presenting them individually to keep the table less cluttered. We used the batch-normalization algorithm (Ioffe and Szegedy, 2015) along with `tanh` nonlinearities. Sampling the RBM was done by performing 200 steps of Gibbs updates for every mini-batch, in accordance with our baselines,using persistent contrastive divergence (PCD) (Tieleman, 2008). We have observed that by having 2 and 20 PCD steps, the performance of our best model on MNIST dataset is deteriorated by 0.35 and 0.12 nats on average, respectively. In order to estimate the log-partition function, $\log Z_\theta$, a GPU implementation of parallel tempering algorithm with bridge sampling was used (Desjardins et al., 2010; Bennett, 1976; Shirts and Chodera, 2008), with a set of parameters to ensure the variance in $\log Z_\theta$ is less than 0.01: $20K$ burn-in steps were followed by $100K$ sweeps, 20 times (runs), with a pilot run to determine the inverse temperatures (such that the replica exchange rates are approximately 0.5).

We underscore several important points regarding Table 1. First, when one sample is used in the training objective ($k = 1$), GumBolt outperforms dVAE and dVAE++ in all cases. This can be due to the efficient use of reparameterization trick and the absence of REINFORCE elements in the structure of GumBolt as opposed to dVAE (Appendix B). Second, the previous models do not apply when $k > 1$. GumBolt allows importance weighted objectives according to Eq. 9. We see that in all cases, by adding more samples to the training objective, the performance of the model is enhanced.

Fig. 2(b) depicts the $k = 20$ estimation of the GumBolt and discrete objectives on the training and validation sets during training. It can be seen that over-fitting does not happen since all the objectives are improving throughout the training. Also, note that the differentiable GumBolt proxy closely follows the non-differentiable discrete objective. Note that the kinks in the learning curves are caused by our stepwise change of the learning rate and is not an artifact of the model.

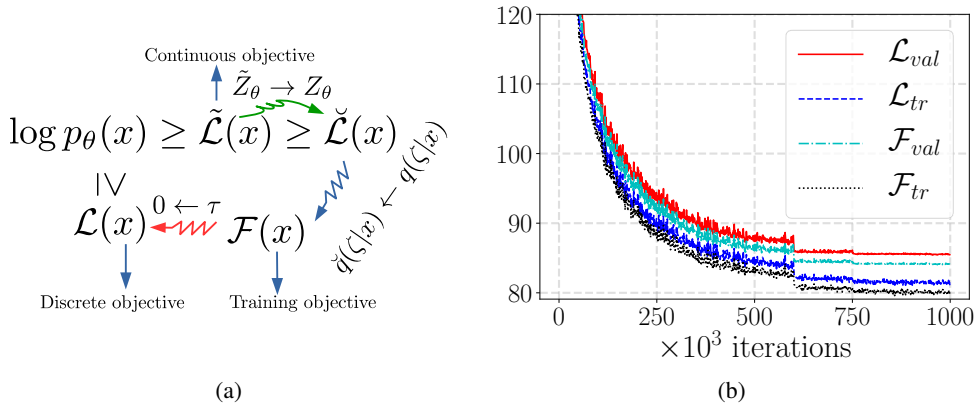

|       | (a) | | (b) |

Figure 2: (a) Relationship between the different objectives. Note that functional dependence on $\theta$ and $\phi$ have been suppressed for brevity. (b) The values of the discrete ($\mathcal{L}$) and GumBolt ($\mathcal{F}$) objectives (with $k = 20$ for all objectives) throughout the training on the training and validation sets (of MNIST dataset) for a GumBolt on $\sim \sim$ structure during $10^6$ iterations of training. The subscripts "val" and "tr" correspond to the validation and training sets, respectively. The abrupt changes are caused by stepwise annealing of the learning rate. This figure signifies that the differentiable GumBolt objective faithfully follows the non-differentiable discrete objective, leading to no overfitting caused by following a wrong objective. We did not use early stopping in our experiments.

## 5.2 Comparison with other discrete models and the importance of powerful priors

If the BM prior is replaced with a factorial Bernoulli distribution, GumBolt transforms into CON-CRETE (Maddison et al., 2016) (when continuous variables are used inside discrete pdfs) and Gumbel-Softmax (Jang et al., 2016). This can be achieved by setting the couplings ($W$) of the BM to zero, and keeping the biases. Since the performance of CONCRETE and Gumbel-Softmax has been extensively compared against other models (Maddison et al., 2016; Jang et al., 2016; Tucker et al., 2017; Grathwohl et al., 2017), we do not repeat these experiments here; we note however that CONCRETE performs favorably to other discrete latent variable models in most cases.

Table 2: Performance of GumBolt (test-set log-likelihood) in the presence and absence of coupling weights. -nW in the second column signifies that the elements of the coupling matrix $W$ are set to zero, *throughout* the training rather than just during evaluation. Removing the weights significantly degrades the performance of GumBolt.

|  |  | GumBolt $k = 20$ | GumBolt-nW $k = 20$ |
|---|---|---|---|
|  | $-$ | **87.45** | 99.94 |
| **MNIST** | $\sim$ | **83.87** | 93.50 |
|  | $\sim\sim$ | **82.75** | 88.01 |
|  | $-$ | **103.69** | 112.21 |
| **OMNIGLOT** | $\sim$ | **100.68** | 107.221 |
|  | $\sim\sim$ | **98.81** | 105.01 |

An interesting question is studying how much of the performance advantage of GumBolt is caused by powerful BM priors. We have studied this in Table 2 by setting the couplings of the BM to $0$ throughout the training (denoted by GumBolt-nW). The GumBolt with couplings significantly outperforms the GumBolt-nW. It was shown in (Vahdat et al., 2018) that dVAE and dVAE++ outperform other models with discrete latent variables (REBAR, RELAX, VIMCO, CONCRETE and Gumbel-Softmax) on the same structure. By outperforming the previuos models with BM priors, our model achieves state-of-the-art performance in the scope of models with discrete latent variables.

Another important question is if some of the improved performance in the presence of BMs can be salvaged in the GumBolt-nW by having more powerful neural nets in the decoder. We observed that by making the decoder's neural nets wider and deeper, the performance of the GumBolt-nW does not improve. This predictably suggests that the increased probabilistic capability of the prior cannot be obtained by simply having a more deterministically powerful decoder.

## 6 Conclusion

In this work, we have proposed the GumBolt that extends the Gumbel trick to Markov random fields and BMs. We have shown that this approach is effective and on the entirety of a wide host of structures outperforms the other models that use BMs in their priors. GumBolt is much simpler than previous models that require marginalization of the discrete variables and achieves state-of-the-art performance on MNIST and OMNIGLOT datasets in the context of models with only discrete variables.

## Footnotes

*Currently at Borealis AI

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
