[Supplementary Material]

# Supplementary material for GumBolt: Extending Gumbel trick to Boltzmann priors

**Amir H. Khoshaman**
D-Wave Systems Inc.
khoshaman@gmail.com

**Mohammad H. Amin**
D-Wave Systems Inc.
Simon Fraser University
mhsamin@dwavesys.com

## A Theorems regarding GumBolt

**Theorem 1.** For any polynomial function $E_\theta(z)$ of $n_z$ binary variables $z \in \{0,1\}^{n_z}$, the extrema of the relaxed function $E_\theta(\zeta)$ with $\zeta \in [0,1]^{n_z}$ reside on the vertices of the hypercube, i.e., $\zeta_{\text{extr}} \in \{0,1\}^{n_z}$.

*Proof.* For a binary variable $z_i$ and an integer $n$, we have

$$z_i^n \equiv \underbrace{z_i \ldots z_i}_{n \text{ times}} = z_i.$$

Therefore, the polynomial function $E_\theta(z)$ can only have linear dependence on $z_i$ and can be written as

$$E_\theta(z) = \sum_i z_i g_{i\theta}(z_{-i}), \tag{1}$$

where $g_{i\theta}(z_{-i})$ is a polynomial function of all $z_{j \neq i}$ with $j < i$, to exclude double-counting. The energy function of a BM is a special case of this equation. The relaxed function will have derivatives

$$\frac{\partial E_\theta(\zeta)}{\partial \zeta_i} = g_\theta(\zeta_{-i}). \tag{2}$$

Due to the linearity of the equation, for nonzero $g_\theta(\zeta_{-i})$ there is always ascent or descent direction for $\zeta_i$, therefore, the extrema will be on the vertices of the hypercube. $\square$

**Theorem 2.** For any polynomial function $E_\theta(z)$ of binary variables $z \in \{0,1\}^{n_z}$, the proxy probability $\breve{p}_\theta(\zeta) \equiv e^{-E_\theta(\zeta)}/Z_\theta$, with $\zeta \in [0,1]^{n_z}$, is a lower bound to the true probability $p_\theta(\zeta) \equiv e^{-E_\theta(\zeta)}/\tilde{Z}_\theta$, i.e., $\breve{p}_\theta(\zeta) \leq p_\theta(\zeta)$, where $Z_\theta \equiv \sum_{\{z\}} e^{-E_\theta(z)}$ and $\tilde{Z}_\theta \equiv \int_{\{\zeta\}} d\zeta e^{-E_\theta(\zeta)}$.

*Proof.* Let $E_{\min}$ be the minimum of $E_\theta(z)$. According to the previous theorem, $E_{\min}$ is also the minimum of $E_\theta(\zeta)$. Therefore

$$\tilde{Z}_\theta = \int_{\{\zeta\}} d\zeta e^{-E_\theta(\zeta)} \leq \int_{\{\zeta\}} d\zeta e^{-E_{\min}} = e^{-E_{\min}} \leq \sum_{\{z\}} e^{-E_\theta(z)} = Z_\theta. \tag{3}$$

Therefore

$$\breve{p}_\theta(\zeta) = \frac{e^{-E_\theta(\zeta)}}{Z_\theta} \leq \frac{e^{-E_\theta(\zeta)}}{\tilde{Z}_\theta} = p_\theta(\zeta). \tag{4}$$

$\square$

## B The equivalence of dVAE and REINFORCE in dealing with the cross-entropy term

In this Appendix, we show that the previous work with a BM prior, dVAE (**?**), is equivalent to REINFORCE when calculating the derivatives of the cross entropy term in the loss function. Note that a discrete variable reparametrized as $z = \mathcal{H}(\rho - (1 - \bar{q}))$ is non-differentiable, due to the discontinuity caused by the Heaviside function. Consider calculating $\nabla_\phi \mathbb{E}_{q_\phi(z|x)} [E_\theta(z)]$, which appears in the gradients of the objective function. The gradients of the coupling terms can be written as:

$$\nabla_\phi \mathbb{E}_{q_\phi(z|x)} \left[ \sum_{i,j}^{n_z} z_i W_{ij} z_j \right] = \mathbb{E}_{\rho \sim \mathcal{U}} \left[ \nabla_\phi \sum_{i,j}^{n_z} z_i(\rho) W_{ij} z_j(\rho) \right]. \tag{5}$$

Using a spike(at 0)-and-exponential relaxation, $r(\zeta_i | z_i)$, *i.e.*,

$$r(\zeta_i | z_i) = \begin{cases} \delta(\zeta_i), & \text{if } z_i = 0 \\ \frac{\exp(-\zeta_i/\tau)}{Z}, & \text{if } z_i = 1, \end{cases} \tag{6}$$

where $Z$ is the normalization constant. It is proved in (**?**) that the derivatives can be calculated as follows:

$$\mathbb{E}_{\rho \sim \mathcal{U}} \left[ \nabla_\phi \sum_{i,j}^{n_z} z_i(\rho) W_{ij} z_j(\rho) \right] = \mathbb{E}_{\rho \sim \mathcal{U}} \left[ \sum_{i,j}^{n_z} \frac{1 - z_i(\rho)}{1 - \bar{q}_i(\rho)} W_{ij} z_j(\rho) \nabla_\phi \bar{q}_i(\rho) \right]. \tag{7}$$

In order to show that this is equivalent to REINFORCE, first consider the spike (at one)-and-exponential distribution:

$$r(\zeta_i | z_i) = \begin{cases} \delta(\zeta_i), & \text{if } z_i = 1 \\ \frac{\exp(-\zeta_i/\tau)}{Z}, & \text{if } z_i = 0, \end{cases} \tag{8}$$

which is equivalent to spike(at 0)-and-exponential relaxation distribution (since there is nothing special about $z = 0$). Using this distribution and the same line of reasoning used in (**?**), the derivatives of the coupling term become:

$$\mathbb{E}_{\rho \sim \mathcal{U}} \left[ \nabla_\phi \sum_{i,j}^{n_z} z_i(\rho) W_{ij} z_j(\rho) \right] = \mathbb{E}_{\rho \sim \mathcal{U}} \left[ \sum_{i,j}^{n_z} \frac{z_i(\rho)}{\bar{q}_i(\rho)} W_{ij} z_j(\rho) \nabla_\phi \bar{q}_i(\rho) \right]. \tag{9}$$

Now consider the REINFORCE trick applied to the coupling term:

$$\nabla_\phi \mathbb{E}_{q_\phi(z|x)} \left[ \sum_{i,j}^{n_z} z_i W_{ij} z_j \right] = \mathbb{E}_{q_\phi(z|x)} \left[ \sum_{i,j}^{n_z} z_i W_{ij} z_j \nabla_\phi \log q_\phi(z_i, z_j | x) \right]. \tag{10}$$

Assuming the general autoregressive encoder, where every $z_i$ depends on all the preceding variables, $z_{<i}$, *i.e.*, we can write

$$\log q_\phi(z_i, z_j | x) = \sum_{\{z_k : k \notin \{i,j\}\}} z_i \log(\bar{q}(z_{<i})) + (1 - z_i) \log(1 - \bar{q}(z_{<i})) + \\ z_j \log(\bar{q}(z_{<j})) + (1 - z_j) \log(1 - \bar{q}(z_{<j})). \tag{11}$$

Replacing this in Eq. 10, and noting that for any binary variable $z_i$ we have $z_i(1 - z_i) = 0$, results in:

$$\begin{aligned}
\nabla_\phi \mathbb{E}_{q_\phi(z|x)} \left[ \sum_{i,j}^{n_z} z_i W_{ij} z_j \right] &= \mathbb{E}_{q_\phi(z|x)} \left[ \sum_{i,j}^{n_z} z_i(z_{<i}) W_{ij} z_j(z_{<j}) \nabla_\phi \log \bar{q}(z_{<i}) \right] \\
&= \mathbb{E}_{\rho \sim \mathcal{U}} \left[ \sum_{i,j}^{n_z} z_i(\rho) W_{ij} z_j(\rho) \nabla_\phi \log \bar{q}_i(\rho) \right] \\
&= \mathbb{E}_{\rho \sim \mathcal{U}} \left[ \sum_{i,j}^{n_z} \frac{z_i(\rho)}{\bar{q}_i(\rho)} W_{ij} z_j(\rho) \nabla_\phi \bar{q}_i(\rho) \right],
\end{aligned} \tag{12}$$

where the last equality is due to the law of unconscious statistician (**?**), *i.e.*, for a given function $f(z)$, we have

$$\mathbb{E}_{\rho \sim \mathcal{U}}\left[f(z(\rho))\right] = \mathbb{E}_{q_\phi(z|x)}\left[f(z)\right].$$

Therefore, dVAE is using REINFORCE when dealing with the cross-entropy terms.