[Reviews · NeurIPS 2018]

Reviewer 1



Well written paper about VAEs with RBM priors. The main innovations are straight forward, seem easy to implement and prove to outperform previous methods. I only have a few comments: - I think there are some minor problems with the proof of Thm 1 (which I think holds nevertheless): In Eq 1 of the proof on the appendix the decomposition is wrong. It should state that $\forall i \exists g, f E=z_ig_i(z_{\backslash i})+f_i(z_{\backslash i})$. Also just argueing that the trace of the Hessian is 0 is not imidiately clear to be sufficient. One can prob argue that for any critical point (unless the function is constant), we can find a directions i such that $g_i(z_{\backslash i})$ is non-zero, and therefore there's a ascend and descend direction in $i$. - Would it also be possible to esitmate $\tilde Z_\theta$ by doing Gibbs sampling with truncated Gaussians instead of replacing it with $Z_\theta$? This should be a tighter bound as far as I understand. - I would replace the term 'indifferentiable' with the more comment 'non-differentiable' - l139: There's smth missing in the first sentence. - I'm not entirely sure I understand the model layout in the experimental section for two stochastic layers. Are those the two componenents in the bipartite graph of an RBM or are there in fact 2 RBMs? This could be explained in a more transparent way. - The labelling of figure 2b should be explained better.

Reviewer 2



# Response to author feedback My thanks to the authors for their detailed feedback to the comments in my review. The clarifications about the GumBolt-nW experiment and further baseline results provided comprehensively addressed my queries about this issue and instil confidence in the authors' conclusion that there is a significant gain to using a RBM prior over simpler factorial choices. After reading the authors responses to my and the other reviews I continue to feel this would be a worthy inclusion in the conference and vote for acceptance. --- # Summary and relation to previous work This submission builds upon a series of previous works proposing methods for including discrete latent variables in variational autoencoder models while retaining the ability to train the models with low-variance reparameterisation trick based gradient estimates of the variational objective by relaxing the discrete latent variables with associated continuous valued variables. In particular it extends the dVAE (Rolfe, 2016) and dVAE++ (Vahdat et al., 2018) models which use a Boltzmann machine (BM) prior on the discrete latent variables by using an analogue of the 'Gumbel trick' relaxation (Maddison et al., 2016; Jang et al., 2016) applied to the BM prior. The resulting model and training approach is argued to be implementationally simpler than the dVAE and dVAE++ approaches while also allowing the use of a tighter importance-weighted variational bound (Burda et al., 2015) which has been found to often improve training performance. The authors empirically demonstrate the efficacy of their proposed 'GumBolt' approach compared to dVAE and dVAE++ in terms of significant improvements in test set log likelihoods on two benchmark binarized image generative model datasets (MNIST and OMNIGLOT) across a range of different architectures. # Evaluation Although the proposed approach is a largely incremental improvement of existing work, I still believe it would be a worthwhile contribution to the conference with the particular synthesis of ideas from various previous pieces of works being used to produce an approach for training VAE models with discrete latent variables which leads to significant gains in empirical performance. The approach is clearly described, with the relevant background on the various ideas being combine (VAEs, Gumbel trick, BMs) succinctly explained and the relationship of the proposed approach to the most closely related previous work (dVAE and dVAE++) well summarised in Figure 1 and the accompanying text. The issue that even with a bipartite connectivity, sampling from BM distributions is challenging is largely glossed over, with block Gibbs often still mixing quite slowly and therefore requiring many iterations to get a close to independent sample - the original dVAE paper illustrated improved test-set log-likelihoods as the number of Gibbs iterations used during training for sampling from the (R)BM prior was increased and it would have been helpful to have at least a short discussion of this issue here and justification for the choice 200 Gibbs updates during training in the experiments. The illustration of the relationship between the various objectives discussed in Figure 2a is helpful, and the inclusion of a plot of the sensible sanity check of whether the proposed relaxed proxy training objective is tracked by the discrete objective is reassuring. In the experiments with results summarised in Table 2 it is unclear if the 'GumBolt-nW' entries correspond to a GumBolt model trained with non-zero weights and then W set to zero *after* training, or a GumBolt model with W set to zero throughout training - could the authors clarify which of these is the case? If the latter then the results do indeed seem to support that the use of a BM prior gives a significant improvement in performance over a simpler factorial prior, though if the removal of the coupling matrix W parameters in the GumBolt-nW case significantly decreases the overall number of parameters in the model it would be interesting to also include an additional baseline in which e.g. a deeper and/or wider networks are used in the decoder model to see if this can counteract the loss of flexibility in the prior. # Minor comments / typos / suggestions L20: 'VAEs tend to ignore the latent code in the presence of powerful decoders' - its not clear exactly what is meant here by 'latent code' and by 'ignore'. It seems implausible for VAE to 'ignore' the latent code in the sense of the latent variables not having a strong influence on the generated output when mapping through the decoder, for example in the case of a Gaussian decoder this would surely correspond to the mean and variance networks outputting fixed values independent of z and so the learned marginal distribution on x being Gaussian. L28: Not clear why that Boltzmann machine represent intractable distributions make them 'ideal for priors in VAEs' L38/39: 'auto-encoding term' is not explicitly defined making it a bit difficult to parse this L51: Citation on this line should use \citet not \citep as directly reference in text L96: 'to facilitates sampling' -> 'to facilitate sampling' - would also be more accurate to make explicit this is MCMC sampling L97: Would be useful to here to state bipartitie connectivity assumed in Boltzmann machine prior often termed restricted Boltzmann machine (as RBM which is used later is never explicitly defined) Eq 5: nitpick but it would more correct for the subscripted distribution density the expectation is indicated to be with respect to to be a product of densities over the sample index $i$ (i.e. joint over independent samples) Eq 6: Looks like there are negatives missing in the expression after the second and third equalities L116: 'probability distribution' -> 'probability density function' would be better here L123: 'theorom' -> 'theorem' Table 1: The wider gap between $k=1$ and $k=5$ GumBolt columns is a bit confusing as makes it appear as if the $k=1$ column is grouped with those to left rather than right. Also need to make cleared in caption and/or main text that these are (presumably) estimated *test-set* log-likelihoods. L199: 'reduced by 0.3' is ambiguous - would be better to be more explicit e.g. is the learning rate multiplied by 0.3 or 1 - 0.3 = 0.7?

Reviewer 3



The work presents a method to train a VAE with binary latent variables and an RBM prior distribution. Starting from the idea of concrete relaxation, the work presents the GumBolt proxy, where the energy function is computed using continuous values while the normalizer is computed using the discrete (binary) values. This allows both the efficient approximation of the normalizer for training the prior and pathway gradient for training the concrete relaxed encoder. Based on the GumBolt proxy, a lower bound of the IWAE objective is proposed to train the VAE, with a further tweak to resolve the discrepancy between pdf and pmf in the limiting case. Empirically, the VAE with an RBM prior significantly outperforms both the DVAE (DVAE++) and the VAE with a factorized Bernoulli prior. Strength: - The proposed method is clean and effective. - A reasonable analysis is provided to justify why the relaxed GumBolt proxy well tracks the discrete RBM prior. Weakness: - The main concern I have is that whether an RBM prior is necessary or the best choice given its complication. To this end, it would be more informative to compare with an auto-regressive prior where each conditional could be relaxed into a concrete distribution (e.g. parameterized by an RNN or (dilated) CNN with comparable parameter size). Since the auto-regressive prior can yield the exact probability, training can be directly performed with the IWAE objective. In terms of the sampling, an auto-regressive prior and an RBM prior could be similarly slow. Overall, I think this is an interesting paper with positive empirical results. However, whether the RBM prior is a necessary or the best choice is not clear yet.